# The Microstructure and Properties of an Al-Mg-0.3Sc Alloy Deposited by Wire Arc Additive Manufacturing

**Lingling Ren [1], Huimin Gu [1],\*, Wei Wang [2], Shuai Wang [1], Chengde Li [1], Zhenbiao Wang [3], Yuchun Zhai [1,3] and Peihua Ma [1]**

[1]  College of Metallurgy, Northeastern University, Shenyang 110000, China; renlingling27@126.com (L.R.); wangshuai106123@126.com (S.W.); lichengde20031698@126.com (C.L.); zhaiyc@smm.neu.edu.cn (Y.Z.); mapeihuadbdx@126.com (P.M.)

[2]  Inner Mongolia Metal Research Institute, Baotou 014000, China; wwneu@hotmail.com

[3]  North East Industrial Materials & Metallurgy Co., Ltd, Fushun 113200, China; zhenbiaowang@hotmail.com

\*  Correspondence: guhm@smm.neu.edu.cn; Tel.: +86-1389-793-7272

**Abstract:** Al-Mg alloys can reach medium strength without a solid solution and quenching treatment, thereby avoiding product distortion caused by quenching, which has attracted the attention of wire arc additive manufacturing (WAAM) researchers. However, the mechanical properties of the WAAM Al-Mg alloy deposits obtained so far are poor. Herein, we describe the preparation of Al-Mg-0.3Sc alloy deposits by WAAM and detail the pores, microstructure, and mechanical properties of the alloy produced in this manner. The results showed that the number and sizes of the pores in WAAM Al-Mg-0.3Sc alloy deposits were equivalent to those in Al-Mg alloy deposits without Sc. The rapid cooling characteristics of the WAAM process make the precipitation morphology, size, and distribution of the primary and secondary $Al_3Sc$ phases unique and effectively improve the mechanical properties of the deposit. A primary $Al_3Sc$ phase less than 3 μm in size was found to precipitate from the WAAM Al-Mg-0.3Sc alloy deposits. The primary $Al_3Sc$ phase refines grains, changes the segregated $β(Mg_2Al_3)$ phase morphology, and ensures that the mechanical properties of horizontal and vertical samples of the deposits are uniform. After heat treatment at 350 °C for 1 h, the WAAM Al-Mg-0.3Sc alloy deposits precipitated a secondary $Al_3Sc$ phase, which was spherical (diameter about 20 nm) and had high dispersity. This phase blocks dislocations and subgrain boundaries, causes a noticeable strengthening effect, and further improves the mechanical properties of the deposits, up to a horizontal samples tensile strength of 415 MPa, a yield strength of 279 MPa, and an elongation of 18.5%, a vertical samples tensile strength of 411 MPa, a yield strength of 279 MPa, and an elongation of 14.5%. This Al-Mg-Sc alloy is expected to be widely used in the WAAM field.

**Keywords:** Al-Mg-0.3Sc alloy; wire arc additive manufacturing; $Al_3Sc$; mechanical properties; heat treatment

## 1. Introduction

Wire arc additive manufacturing (WAAM) has exhibited unique advantages in large-scale component manufacturing due to its high material utilization rate, high deposition rate, low production and equipment cost, and high equipment flexibility and scalability [1–3]. WAAM of aluminum alloys have been widely investigated by researchers. At present, the alloys studied mainly include the Al-Cu alloy, Al-Cu-Mg alloy, and Al-Si-Mg alloy [4–7]; excellent mechanical properties have been obtained from these alloys. However, these alloys require solution quenching and aging treatment. The quenching process easily leads to product distortion, seriously reducing the yield. Al-Mg alloys can reach medium strength (tensile strength 310–350 MPa [8]) without a solid solution and quenching

treatment, thus WAAM of Al-Mg alloy parts can avoid product distortion caused by quenching and can provide rapid manufacturing. Therefore, Al-Mg alloys are more suitable for WAAM than alloys that require hardening via a solid solution and quenching treatment.

The mechanical properties of the WAAM Al-Mg alloy deposits produced in previous studies are relatively poor. For example, Horgar et al. reported tensile and yield strengths of 293 MPa and 145 MPa, respectively, for WAAM AA5183 aluminum alloy deposits, while Geng et al. used gas tungsten arc welding to additively manufacture 5A06 alloy deposits, which resulted in lower tensile and yield strengths of 273 MPa and 124 MPa, respectively [9,10].

Sc is regarded as the most effective alloying element for aluminum alloys [11]; it can form $Al_3Sc$ particles in an aluminum alloy. Further, this primary $Al_3Sc$ phase can be used as heterogeneous nucleation center to refine grains. The secondary $Al_3Sc$ phase is small and dispersed, which blocks dislocations and grain boundaries and strengthens the alloy. Consequently, Sc significantly improves the mechanical properties of aluminum alloys [12–15]. The morphology and distribution of the segregated phases of primary and secondary $Al_3Sc$ are related to the cooling rate of the melt. The higher the cooling rate of the melt, the smaller the size of the segregated phase of the primary $Al_3Sc$ and the more significant the grain refinement effect [16,17]. The higher the cooling rate of the melt, the more Sc remains in the solid solution, which provides favorable conditions for the precipitation of a large number of secondary $Al_3Sc$ phases in subsequent thermal processing. The advantages of the primary $Al_3Sc$ and secondary $Al_3Sc$ precipitated phases can be fully realized due to the fast cooling rate of the WAAM process. At present, no research has been published on the WAAM Al-Mg-Sc alloy.

In this paper, cold metal transfer (CMT)-based WAAM was applied to produce Al-Mg-0.3Sc alloy deposits, where a CMT advanced (CMT-ADV) process was used. The microstructure and mechanical properties of this alloy were studied and compared with alloy deposits devoid of Sc. The purposes of this study were to investigate the role of Sc during the preparation of the Al-Mg alloy by WAAM, improve the mechanical properties of the deposits, and provide a foundation for the development of WAAM Al-Mg alloys.

## 2. Materials and Methods

The Al-Mg alloy wires used in this study were provided by North East Industrial Materials and Metallurgy Co., Ltd. (Fushun, China), and were 1.2 mm in diameter. The additive manufacturing system includes a Fronius Advance 4000 arc welding power supply and an ABB 1410 welding robot. The device is shown in Figure 1.

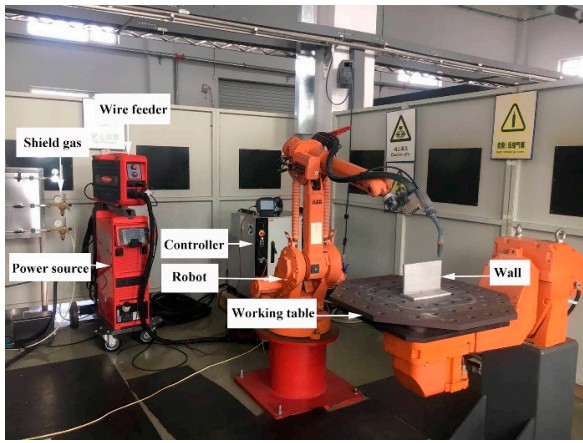

**Figure 1.** The cold metal transfer wire arc additive manufacturing (CMT-WAAM) system.

In this experiment, two kinds of welding wire were used to prepare the deposits: Al-Mg alloy and Al-Mg-0.3Sc alloy wires. The wire compositions are listed in Table 1, and the process parameters are provided in Table 2. The deposits were 200 mm × 150 mm × (5.5–6) mm in size. The Al-Mg-0.3Sc alloy deposits were heat treated at 350 °C for 1 h and followed by air cooling to room temperature.

**Table 1.** Chemical compositions of the welding wires (mass%).

| Alloys | Si | Fe | Mn | Mg | Ti | Sc | Al |
|---|---|---|---|---|---|---|---|
| Al-Mg | 0.0191 | 0.0889 | 0.745 | 6.32 | 0.137 | - | balance |
| Al-Mg-0.3Sc | 0.0179 | 0.0905 | 0.724 | 6.39 | 0.134 | 0.28 | balance |

**Table 2.** Process parameters for the Al-Mg and Al-Mg-0.3Sc alloys.

| Process Parameters | |
|---|---|
| Current | 90 A |
| Arc voltage | 10 V |
| Travel speed | 8 mm/s |
| Wire feed speed | 5.5 mm/min |
| Argon(99.999%) flow rate | 25 L/min |
| Interpass temperature | 60–80 °C |

The additive manufacturing process is shown in Figure 2a, where the *x*-axis corresponds to the front side of the deposits, the *y*-axis is the moving direction of the torch, and the *z*-axis is the deposition direction of the deposits. Three tensile test samples perpendicular to the deposition direction (horizontal samples) and three tensile test samples parallel to the deposition direction (vertical sample) were extracted from the deposits. The specimen's dimensions are reported in Figure 2b. Tensile tests were performed at room temperature using an electro-mechanical universal testing machine. An ICAP7400 plasma spectrometer (Thermo Scientific, Massachusetts, Waltham, American) was used for Al-Mg alloy wire detection. The metallographic specimens (from the location 3 of the deposit shown in Figure 2a) were ground and polished use 0.5 μm diamond paste to a mirror finish and then etched in a mixed acid reagent containing 1 vol% HF, 1.5 vol% HCl, and 2.5 vol% $HNO_3$, with the balance consisting of $H_2O$. The microstructures were characterized using a LEICA MEF4M optical microscope (OM, Astria LEICA, weztlar, Germany), a QUANTA FEG 250 scanning electron microscopy (SEM, Zeiss, London, UK), and a FEI Titan Themis transmission electron microscope (TEM, Titan Themis, New York, NY, USA). The micro-area composition was measured using energy dispersive spectrometry (EDS, Zeiss, London, UK).

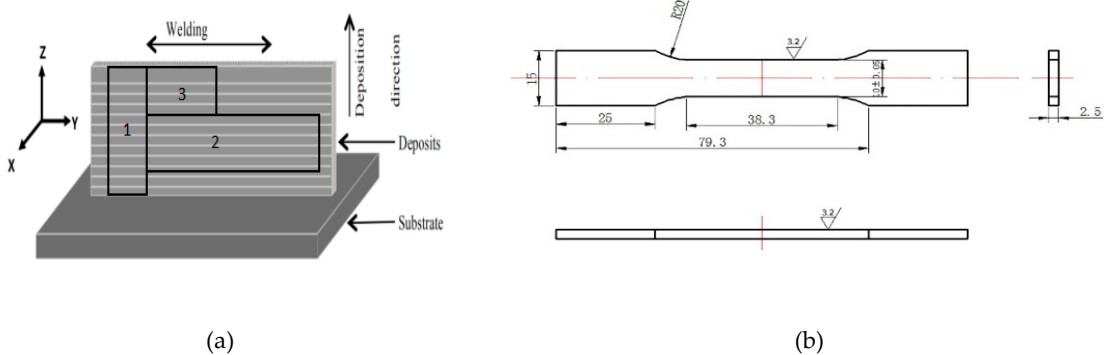

(a)            (b)

**Figure 2.** (**a**) definition of coordinates for wall and (**b**) dimensions of the specimen (the units for the specimen's dimensions are mm).

## 3. Results and Discussion

### 3.1. Pores

Porosity can cause stress concentration, which is an important defect that affects the performance of aluminum alloys and can cause fractures. Therefore, it is necessary to investigate porosity. Figure 3 shows the pores of the as-deposited Al-Mg alloy and Al-Mg-0.3Sc alloy deposits. It can be seen that the pores of both deposits are round, with a diameter of about 40 μm and a dispersed distribution. It is widely believed that hydrogen is the main cause of the porosity in aluminum alloy. The CMT + ADV process has a low heat input and a high solidification rate, so the bubbles have no time to nucleate and grow [18]. This indicates that the addition of 0.3 wt % Sc will not affect the formation of pores under this process.

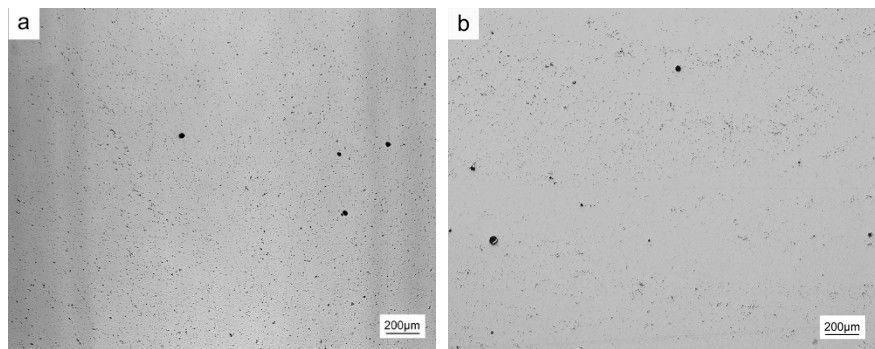

**Figure 3.** Optically observed porosity for the WAAM-fabricated (**a**) Al-Mg alloy and (**b**) Al-Mg-0.3Sc alloy.

### 3.2. Primary Al$_3$Sc Phase

The primary Al$_3$Sc phase is a very important segregated phase in Al-Mg-Sc alloys. The primary Al$_3$Sc phase forms an L1$_2$-type lattice similar in size to the matrix Al, with a degree of mismatch of only 1.5% and coherency with the α(Al) matrix [19,20]. The Al$_3$Sc phase, with its high melting point (1320 °C), precipitates preferentially during alloy solidification to produce a stable, evenly distributed structure. These characteristics ensure that the primary Al$_3$Sc phase provides good heterogeneous nucleation particles that refine the aluminum alloy grains [12–15]. However, the effect of the primary Al$_3$Sc phase is directly related to the cooling rate of the melt. Hyde et al. studied the influence of cooling rate on the morphology of the segregated phase of primary Al$_3$Sc in Al-Sc alloys. Under an as-cast condition, as the cooling rate increased from 1 to 1000 K/s, the average size of the primary Al$_3$Sc phase decreased from 400 to 20 μm, and the morphology also showed great differences [17]. Compared with the casting, WAAM has a higher cooling rate, and the morphology and fine grain effect of the primary Al$_3$Sc phase are unique.

Figure 4 shows the metallographic structures of the Al-Mg alloy and Al-Mg-0.3Sc alloy deposits, which reveal that the deposit grains were strongly refined by the addition of Sc due to the precipitation of the primary Al$_3$Sc phase during the WAAM process. The average grain size decreased from about 30 to about 15 μm. Figure 5 shows the morphology of the segregated phase of the deposits. Figure 5a,c shows the segregated phase morphology of the Al-Mg alloy and Al-Mg-0.3Sc alloy deposit. According to spectrogram d1 and spectrogram d2, the structure and image of the aluminum alloy [21] and the binary phase diagrams of Al-Mg and Al-Sc show that the two precipitated phases are the β(Mg$_2$Al$_3$) phase and Al$_3$Sc phase, respectively. Figure 5a indicating that the segregated phase is mainly the β(Mg$_2$Al$_3$) phase, which is large and continuously segregated along the grain boundary. A comparison between Figure 5a,b shows that after Sc is added, the size of the segregated phase decreases, and the segregated phase along the grain boundary changes from continuous to non-continuous. Figure 5c shows the morphology of the primary Al$_3$Sc and β(Mg$_2$Al$_3$) phase in the deposit of Al-Mg-0.3Sc alloy. It can be seen that the primary Al$_3$Sc phase is bright white and in the shape of a "cluster". Its size is less than 3

μm, which is much smaller than the size of the primary Al$_3$Sc phase in Al-Mg-Sc alloy formed by the cast [15,16]. The primary Al$_3$Sc phase is segregated with the β(Mg$_2$Al$_3$) phase.

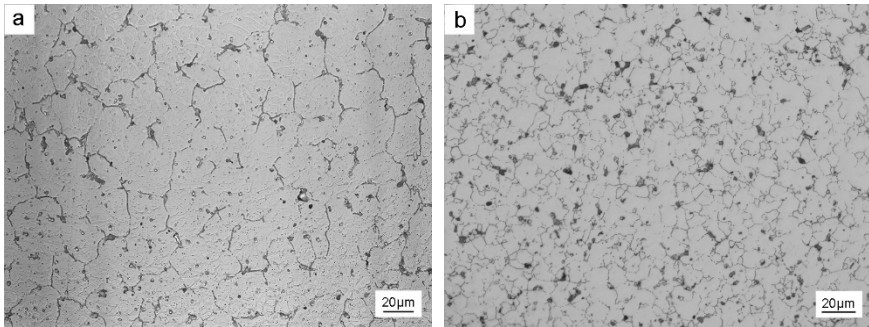

**Figure 4.** Images showing the metallographic structures of (**a**) Al-Mg alloy deposits and (**b**) Al-Mg-0.3Sc alloy deposits.

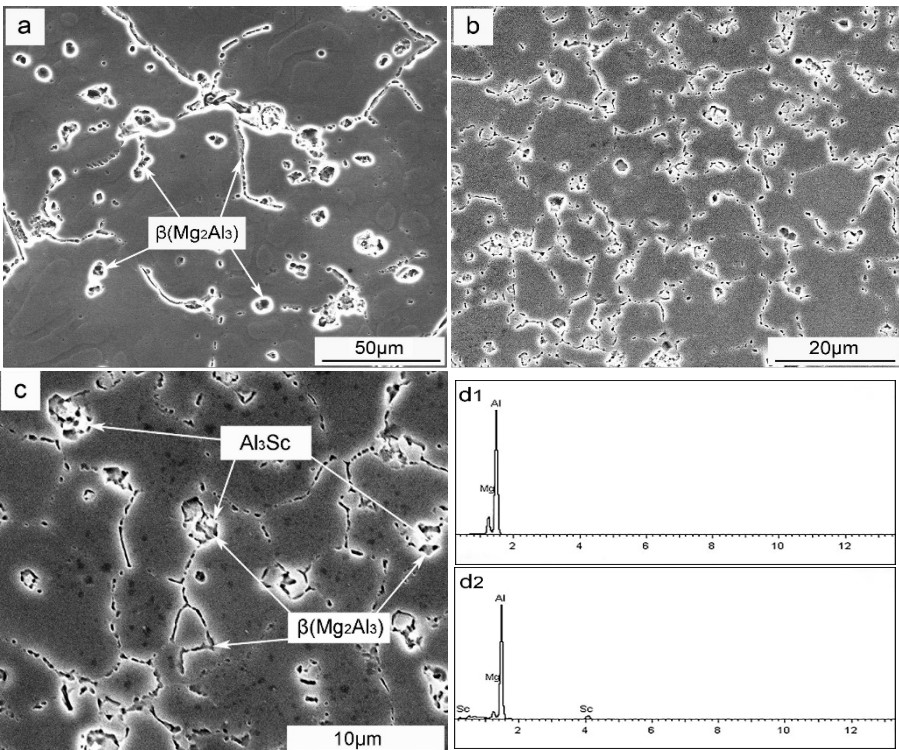

**Figure 5.** SEM images of (**a**) Al-Mg alloy deposits and (**b**,**c**) Al-Mg-0.3Sc alloy deposits. (**d1**) Energy dispersive spectrometry (EDS) spectra of the primary Al$_3$Sc phase and (**d2**) EDS spectra of the Mg$_2$Al$_3$ phase.

According to the analysis of the above results, due to the fast cooling of the WAAM process, the primary Al$_3$Sc phase with a size of less than 3 μm was obtained. The primary Al$_3$Sc phase with this size as the center of the heterogeneous nucleation had an obvious fine α(Al) grains effect. The primary Al$_3$Sc phase not only created fine α(Al) grains but also greatly changed the size and morphology of the segregated β(Mg$_2$Al$_3$) phase. The solid solubility of Mg in aluminum was 1.9 wt % at room temperature. When the Mg content was greater than 3 wt %, the β(Mg$_2$Al$_3$) phase segregated along the grain boundary. The β(Mg$_2$Al$_3$) phase had a face-centered cube and was very brittle at room temperature. The large and continuous segregated β(Mg$_2$Al$_3$) phase will reduce the mechanical properties of the alloy and increase the stress corrosion sensitivity [21]. As can be seen from Figure 5c, primary Al$_3$Sc was mostly accompanied with β(Mg$_2$Al$_3$), which blocked the continuous segregation and growth of the β(Mg$_2$Al$_3$) phase, making the size of the β(Mg$_2$Al$_3$) phase smaller and discontinuous

along the grain boundary. It can be seen that the Al-Mg-0.3Sc alloy prepared by WAAM could form a fine primary $Al_3Sc$ phase to play the role of fine grain strengthening, and could also change the morphology of the segregated phase to increase the mechanical properties of the alloy.

### 3.3. Secondary $Al_3Sc$ Phase

The secondary $Al_3Sc$ phase is another very important precipitated phase in aluminum alloy containing Sc and can be precipitated in subsequent heat treatment processing. These secondary $Al_3Sc$ particles immobilize dislocations and subgrain boundaries and greatly increase the shear stress required for dislocation slippage, which strengthens the alloy.

Figure 6 shows TEM and HRTEM images of the Al-Mg-0.3Sc alloy deposits after 350 °C/1 h heat treatment, which revealed that the secondary $Al_3Sc$ phase that precipitated from the matrix was small and spherical, with well-dispersed particles with diameters of about 20 nm that were completely coherent with the $\alpha(Al)$ matrix (as shown in Figure 6b). These characteristics ensure that the secondary $Al_3Sc$ phase had a good precipitation strengthening effect.

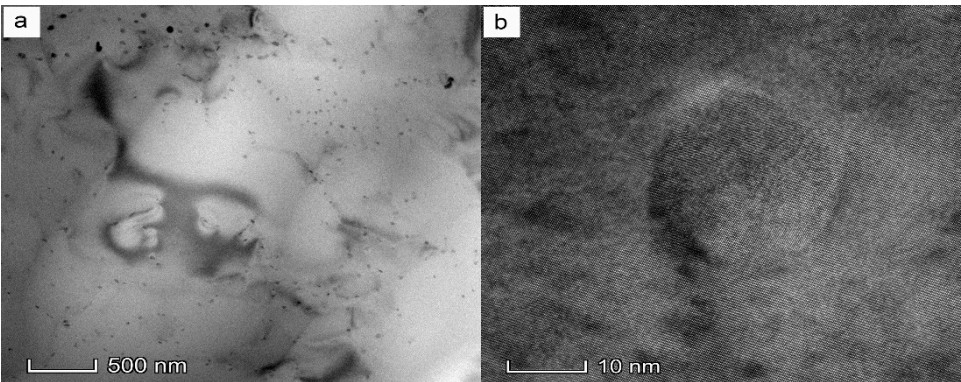

**Figure 6.** (**a**) TEM and (**b**) HRTEM images of the Al-Mg-0.3Sc alloy deposits after heat treatment.

It can be seen that, in combination with the characteristics of fast melting and quick solidification in the WAAM process, the addition of Sc to the Al-Mg alloy could make uniformly solid Sc dissolve in the aluminum matrix, providing favorable conditions for the precipitation of a large number of dispersed secondary $Al_3Sc$ phases.

### 3.4. Mechanical Properties and Fracture Morphology

Figure 7 displays the mechanical properties of horizontal (H) and vertical (V) samples of the deposits, and shows that the mechanical properties of the horizontal and vertical samples of the Al-Mg alloy deposits were greatly different (1#). The mechanical properties of the Al-Mg-0.3Sc alloy deposits (2#) were significantly superior to those of the Al-Mg alloy and were uniform in both the horizontal and vertical directions. After heat treatment at 350 °C/1 h (3#), the strength of the Al-Mg-0.3Sc alloy deposit was further increased, and elongation was reduced. The horizontal samples had a tensile strength, yield strength, and elongation of 415 MPa, 279 MPa, and 18.5%, respectively; the vertical samples' tensile strength, yield strength, and elongation were 411 MPa, 279 MPa, and 14.5%, respectively.

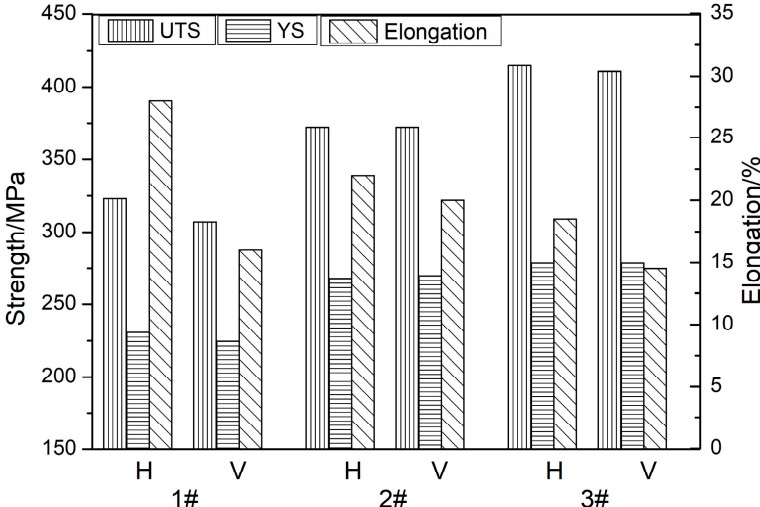

**Figure 7.** Mechanical properties of the deposits. 1#: Al-Mg alloy deposits; 2#: Al-Mg-0.3Sc alloy deposits; and 3#: Al-Mg-0.3Sc alloy deposits after heating at 350 °C for 1 h. UTS = ultimate tensile strength; YS = yield strength.

Figure 8a,b, respectively, shows the vertical and horizontal fracture morphology of the Al-Mg alloy deposit, respectively. It can be seen that there are obvious dimples in the fracture, indicating a ductile fracture. Figure 8c,d shows the vertical and horizontal fracture morphology of the Al-Mg-0.3Sc alloy deposit, respectively. Compared with the fracture morphology of the deposit of the Al-Mg alloy, the dimples were small and had an obvious "tear pattern", which is a sign of high strength and high toughness. Figure 8e,f, respectively, show the vertical and horizontal fracture morphology of the Al-Mg-0.3Sc alloy deposit after heat treatment at 350 °C/1 h, with dimples becoming shallower indicating decreased plasticity.

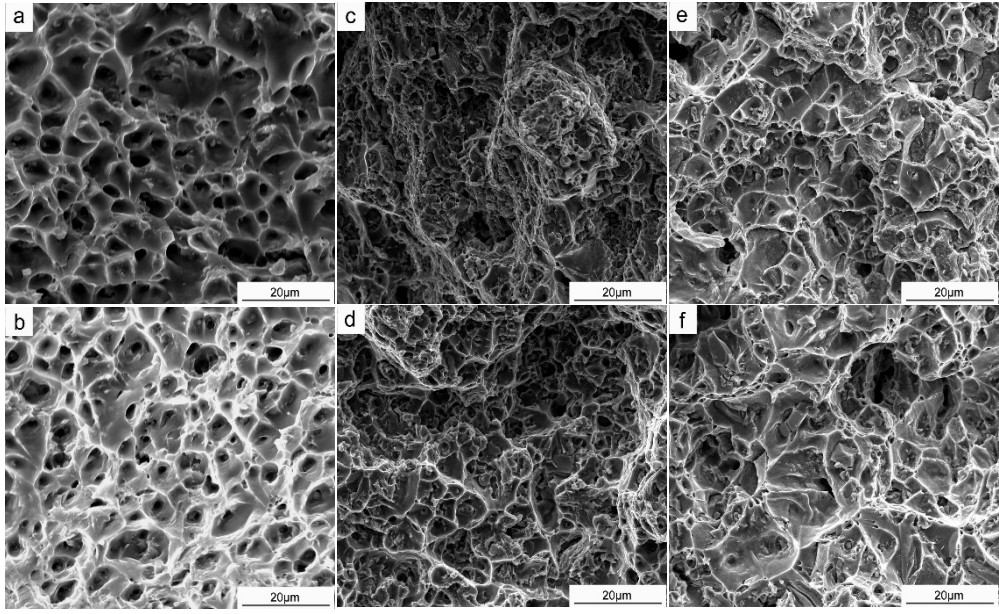

**Figure 8.** Fracture morphologies of the tensile samples of (**a**,**b**) Al-Mg alloy deposits, (**c**,**d**) as-deposited Al-Mg-0.3Sc alloy, and (**e**,**f**) Al-Mg-0.3Sc alloy deposits after heating at 350 °C for 1 h. Where a, c, and e are vertical, and b, d, and f are horizontal.

## 4. Conclusions and Prospects

(1) A primary $Al_3Sc$ phase (<3 μm in size) segregated from WAAM Al-Mg-0.3Sc alloy deposits. This phase refined grains and improved the morphology of the $\beta(Mg_2Al_3)$ phase, which greatly improved the mechanical properties of the deposits and ensured horizontal and vertical uniformity.

(2) A secondary $Al_3Sc$ phase of 350 °C/1 h was precipitated from the WAAM Al-Mg-0.3Sc alloy deposits. This phase contained well-dispersed spherical particles with diameters of about 20 nm; these particles blocked dislocations and subgrain boundaries and further improved the mechanical properties of the deposits, achieving a horizontal samples tensile strength of 415 MPa, a yield strength of 279 MPa, and an elongation of 18.5%, a vertical samples tensile strength of 411 MPa, a yield strength of 279 MPa, and an elongation of 14.5%.

The method of the WAAM Al-Mg-Sc alloy gave full play to the technological characteristics of WAAM and the role of Sc in Al-Mg alloy, and excellent mechanical properties were obtained. It is an important breakthrough in the WAAM Al-Mg alloy field. In future work, the Sc content, form of the arc heat source, process parameters, and post-treatment system will be further studied to promote the engineering application of WAAM Al-Mg-Sc alloys.

**Author Contributions:** Conceptualization, Y.Z. and P.M.; methodology, W.W.; software, C.L. and Z.W.; validation, L.R., S.W. and H.G.; formal analysis, L.R.; investigation, L.R.; resources, W.W.; data curation, S.W., H.G.; writing—original draft preparation, L.R.; writing—review and editing, L.R.; visualization, L.R. and C.L.; supervision, W.W.; project administration, W.W. All authors have read and agreed to the published version of the manuscript.

**Funding:** This research received no external funding.

**Acknowledgments:** The authors thank the National Key Research and Development Plan "Additive Manufacturing and Laser Manufacturing" Key Project (project No. 2018yfb1106300-5) for support.

**Conflicts of Interest:** The authors declare no conflict of interest.

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
