# Peer review of "The Microstructure and Properties of an Al-Mg-0.3Sc Alloy Deposited by Wire Arc Additive Manufacturing"

_metals, doi:10.3390/met10030320_

Round 1

Reviewer 1 Report

In this work two Al alloys obtained with waam technique were compared. More precisely the second one was an Al-Mg-Sc alloy, which after heat treatment, showed better mechanical properties with respect to a simpler Al-Mg alloy. The paper structure is solid and conclusions are clear, despite this some doubts should be clarified. I suggest the publication of this paper after answaring the following questions

In line 41-47, the authors declare that Al-Mg alloys suffer of quenching deformation. Since the material has already good mechanical properties after the waam process, there's no need to heat treat the samples thus avoiding the distortion. how is it possible to avoid deformation during the printig process itself? i think that the cooling rate during solidification could lead to residual stresses similar to those brought by the quenching  in the material. Please add some paragraphs dealing with processability of Al-Mg alloys via waam.

line 60 61 the faster the more...some word is missed do you mean the faster the cooling rate the more Sc remains in solid solution? please revise.

line 80-81 is this treatment a stress relief annealing? please give more details. only the deposits made of Al-Mg-Sc? why this treatment was not performed also on Al-Mg deposits?

line 88 which components? those in tab 1? please clarify

line 106 the CMT+ADV process these acronyms were never used or explained  please revise.

Figure 3 the contrast of the two picture is too different. in my opinion Al-Mg shows a higher degree of porosity. Did you measure the porosity of the sample with image analisis? some quantitative data should be used to validate your statement.

line 125 which particles have changed size? again fig 4 shows two picture with totally different contrast. i think grazing light was used on the left while direct lighting was used on the right. try to reduce these differences with a photo editor software.

I desume that in sample a fig 5 all the precipitates are Mg2Al3 particles but you should discuss it in the text and indicating it in the pictures.

139-142 is not clear please revise.

line 164 ok, so the previously mentioned HT was the ageing treatment? how temperature and time were decided?

Reviewer 2 Report

The manuscript entitled “Microstructure and properties of an Al-Mg-0.3Sc alloy deposited by wire arc additive manufacturing” presents an experimental study of an Al-Mg-0.3Sc alloy produced by wire arc additive manufacturing (WAAM), including its comparison with a WAAM Al-Mg alloy. For this purpose, authors produced Al-based specimens with WAAM and used the optical microscopy, scanning electron microscopy, energy-dispersive X-ray spectroscopy, transmission electron microscopy as well as the electro-mechanical universal testing machine to investigate the microstructure and mechanical properties of as-built WAAM Al-Mg, Al-Mg-0.3Sc alloys as well as heat-treated WAAM Al-Mg-0.3Sc.

Despite the proper experimental research, the manuscript in its current state cannot be published due to the low readability. The work should be clearly communicated. The manuscript needs a thorough revision by a native English proofreader as it contains multiple grammar and language issues. In addition, some of the conclusions should be justified with evidence provided.

Below I give my comments and remarks:

Throughout the manuscript: Do you mean a particular alloy when writing “Al-Mg alloy”. If not, I would recommend writing “alloys” in the plural. Throughout the manuscript: WAAM is wire arc additive manufacturing. Please avoid such wordings as “WAAM process manufacturing” (i.e., “wire arc additive manufacturing process manufacturing”), “WAAM forming process” (i.e., “wire arc additive manufacturing forming process”) or “WAAM additive manufacturing” (i.e., “wire arc additive manufacturing additive manufacturing”). Introduction: Page 1, lines 21-23: I understand the general sense of the sentence but would encourage further editing for better readability, in particular the part describing a secondary Al3Sc phase. Please clarify what you mean under “discrete accumulation” as well as “accumulation” and “preparation” of what is meant here (p. 1, lines 33-35:). I would recommend reformulating. You start writing about the advantages of WAAM but detail a disadvantage of casting and welding in the same sentence. In addition, please explain how flexibility and microstructural consistency would help in the development of the Al-Mg alloys (p. 1, lines 35-38). Please clarify the term “quenching deformation” (p. 1, line 38). It might help the reader if you explained how the “quenching deformation” is related to WAAM. Page 2, lines 45-47: I would encourage further editing for better readability, in particular “wire arc additive manufacturing process manufacturing”. In addition, “more suitable” than what? Page 2, lines 60-62: Please improve readability. “the more Sc the solid” - ? It might help the reader if you explained the abbreviations CMT (legend of Fig. 1, p. 1) and ADV (p. 4, line 106). Usually we write “Figure/Fig.” with the number from the capital F (p. 2, line 75, p. 5, lines 148-149). Results and discussion: The statements in lines 99-101 (p. 3) are of general knowledge, and, in my opinion, do not require any references. In my opinion, it is hard to conclude on the equivalency of the number and size of pores in two alloys based on a single image (p. 3, lines 102-103). I would suggest providing some statistics. Please explain why you refer to the article by Anyalebechi [18] when discussing primary Al3Sc phase and its role in grain refinement (p. 4, lines 115-116). Unfortunately, I could not find anything on Al3Sc in [18]. Please clarify why you write about the work of Hyde et al. (p. 4, lines 117-120) but also refer to the article by Du et al. [15]. In addition, it might help the reader if you mentioned in the link to [15] the language in which the article has been written as it is not English. You repeat the same fact in lines 125-126 and 127-128 (p. 4). I would encourage further editing of lines 139-143 (p. 5) as I can hardly understand what you mean here. How can the primary Al3Sc phase “make the size and morphology of the precipitated phase change greatly”? Please clarify why the “continuous precipitation and growth of β(Mg2Al3) phase are interrupted” (p. 5, lines 148-149). “Preparation can form a Al3Sc phase” – ? (p. 5, lines 150-151). It might help the reader if you clarified how the secondary Al3Sc phase is prone to deviation as well as to deviation from what it is prone (p. 5, lines 158-162). Lines 180-182 (p. 6) appear to be an incomplete sentence. Consider rewriting the sentence. It is not clear whether you write about Al-Mg alloy or Al-Mg-0.3Sc alloy in lines 189-191 (p. 7). Without statistics, I find the conclusion (1) (p. 7, lines 195-196) extraneous. All references should be presented in the same format.

Round 2

Reviewer 2 Report

I very much appreciate the authors’ efforts to meet the referees’ requirements. However, I am still convinced that the paper cannot be accepted due to the basic lack of readability. The manuscript still contains language issues. In addition, some of the conclusions should be justified with evidence provided.

Below I give my comments and remarks:

Abstract: Line 14: WAAM as an abbreviation was already mentioned in line 12. Could you please clarify what the “accumulation body” is (line 19)? I would suggest reformulating “horizontal and vertical mechanical properties” (line 22). Results and discussion: Page 4, line 118: Please clarify what “the degree of the fine grain” is. I would encourage further editing of lines 138-139 (p. 5) as I can hardly understand what you mean here. The same is valid for lines 139-141, 145-148, as I am concerned with the lack of readability. The cause-effect relation is not so clear. 8, the legend: this is the part of the sentence (lines 181-182). Conclusions and prospects: In my opinion, it is hard to conclude on the noneffect of the Sc addition on pores in two alloys based on a single image (Fig. 3). I would suggest providing some statistics. Without statistics, I find the conclusion (1) (p. 7, line 192) extraneous. It looks like a secondary Al3Sc phase is heat-treated at 350 °C (lines 197-198). Please reformulate. I would encourage further editing of lines 203-205 as I can hardly understand what you mean here. All references should be presented in the same format.

Round 3
